# The Identification and Cytotoxic Evaluation of Nutmeg (*Myristica fragrans* Houtt.) and Its Substituents

**DOI:** 10.3390/foods12234211

**Published:** 2023-11-22

**Authors:** Suthiwat Khamnuan, Ampai Phrutivorapongkul, Pornsiri Pitchakarn, Pensiri Buacheen, Jirarat Karinchai, Chuda Chittasupho, Mingkwan Na Takuathung, Tinnakorn Theansungnoen, Kannika Thongkhao, Aekkhaluck Intharuksa

**Affiliations:** 1Department of Pharmaceutical Sciences, Faculty of Pharmacy, Chiang Mai University, Suthep, Mueang, Chiang Mai 50200, Thailand; suthiwat_khamnuan@cmu.ac.th (S.K.); ampai.phrutiv@cmu.ac.th (A.P.); chuda.c@cmu.ac.th (C.C.); 2Department of Biochemistry, Faculty of Medicine, Chiang Mai University, Chiang Mai 50200, Thailand; pornsiri.p@cmu.ac.th (P.P.); pensiri_bua@cmu.ac.th (P.B.); jirarat.ka@cmu.ac.th (J.K.); 3Department of Pharmacology, Faculty of Medicine, Chiang Mai University, Chiang Mai 50200, Thailand; mingkwan.n@cmu.ac.th; 4Clinical Research Center for Food and Herbal Product Trials and Development (CR-FAH), Faculty of Medicine, Chiang Mai University, Chiang Mai 50200, Thailand; 5Green Cosmetic Technology Research Group, School of Cosmetic Science, Mae Fah Luang University, Chiang Rai 57100, Thailand; tinnakorn.the@mfu.ac.th; 6School of Languages and General Education, Walailak University, Nakhon Si Thammarat 80160, Thailand; kannika.to@wu.ac.th

**Keywords:** adulterant, chemical constituent, cytotoxicity, DNA barcode, mace, *Myristica*, nutmeg, spice, substituent

## Abstract

The aril and seed of nutmeg, *Myristica fragrans* Houtt. (Myristicaceae), hold significant value in various industries globally. Our preliminary research found two morphological variations: a globose shape and an oval shape. Due to these different characteristics, the safety of consumers is of primary concern. Thus, authentication and comparative pharmacological and toxicity analyses are necessary. In this study, pharmacognostic and advanced phytochemical analyses, DNA barcoding, cytotoxicity, and the anti-nitric oxide production of commercial Thai nutmeg were examined. Via morphologic examinations and TLC fingerprinting, all the sampled aril and seed were categorized into globose and oval-shaped groups. The results of HPLC, GC-MS, and LC-MS/MS experiments revealed distinct differences between these groups. The DNA barcoding of the *trn*H-*psb*A region using the BLAST method and neighbor-joining tree analyses confirmed the globose nutmeg as *M. fragrans* and the oval-shaped variant as *M. argentea*. A comparison was then carried out between the potential toxicity and anti-inflammatory capabilities of *M. fragrans* and *M. argentea*. Cytotoxicity tests on HaCaT, 3T3-L1, Caco-2, HEK293, and RAW264.7 were performed using both methanolic extracts and volatile oil from the arils and seeds of both species. This study concludes that blending or substituting these two species maintains their therapeutic integrity without posing safety concerns.

## 1. Introduction

*Myristica fragrans* Houtt. (Myristicaceae), or nutmeg, is native to Moluccas, Indonesia, and cultivated in various tropical countries such as Indonesia, India, Sri Lanka, Malaysia, and Thailand [1]. Nutmeg comes from the kernel of the dried ripe seed of *M. fragrans*, whereas mace or macis is derived from the dried aril of the ripe fruit. Both arils and seeds have been widely used in cooking and herbal medicine. *M. fragrans* is an evergreen tree with elliptic or elliptic-lanceolate leathery leaves and urceolate or bell-shaped flowers (Figure 1) [2,3,4]. The nutmeg plant yields 5 to 15% of all volatile oils, which are produced from seeds [5] and used in the perfumery and pharmaceutical industries [6,7]. Both aril and seed crude drugs have been documented in pharmacopeias worldwide [1], and their properties have been applied to treat various diseases [8]. They possess various medicinal properties, including anti-dysenteric, antibacterial, antifungal, anti-oxidant, anti-inflammatory, antithrombotic, aphrodisiac, hepatoprotective, neuro-stimulant, and stomachic activities [9,10]. Their active constituents have been reported, such as α-pinene, elemicin, 4-terpineol, myristicin, eugenol, safrole, and linalool, which are responsible for their pharmacological activities [5,11]. Among these active compounds, myristicin is the main component found in the volatile oil from nutmeg. It can provide potential therapeutic targets that have potential therapeutic applications, particularly in addressing athrosclerosis [12]. However, myristicin has psychoactive properties and can be toxic in large amounts [13,14]. Consequently, it is imperative to exercise caution when it comes to consumption.

Aside from its potential toxicity, the increasing demand for nutmeg may also give rise to situations involving the adulteration of this spice. Currently, major nutmeg-producing countries include Indonesia, India, Sri Lanka, and Grenada, with total global production at around 10,000 to 12,000 ton/year [15] and the European Union is a significant importer of nutmeg products [16]. Previous studies have reported that there are numerous cases of fraudulent nutmeg; for instance, nutmeg becomes spoiled when exposed to materials such as pericarp and shell powder, as well as spent powder (leftover after oil extraction) [15]. Moreover, it can be adulterated with coffee husks [17,18]; *Syzygium aromaticum* (L.) Merr. & L.M.Perry [19] and its related *Myristica* species, namely *Myristica malabarica* Lam. [15,20], *Myristica succedanea* Blume [21], and *Myristica argentea* Warb. [21]. However, discerning processed ingredients from their adulterated counterparts poses challenges for species identification with the naked eye [22]. Therefore, the identification, authentication, and quality control of nutmeg products play pivotal roles in protecting consumer safety, rights, and interests [22].

Various identification methods have been proposed due to the presence of different adulterants in nutmeg products. The Thai Herbal Pharmacopoeia 2022 (THP 2022) recommends macroscopic and microscopic techniques for morphological examination, while thin-layer chromatography (TLC) is recommended for determining the phytochemical components in nutmeg arils and seeds [23]. Highly sensitive methods, such as DNA barcoding, PCR-based techniques for identifying genetic variations (RAPD), and sequence characterized amplified regions (SCARs) have been utilized to confirm nutmeg adulteration [20]. Hyperspectral imaging (HIS) can distinguish between *M. fragrans* powder and adulterants. Fourier transform near-infrared (FT-NIR) spectroscopy rapidly detects ground nutmeg, which is often mixed with nutmeg shells [24]. Gas chromatography–mass spectrometry (GC-MS) is employed to analyze the phytochemical constituents of nutmeg seeds [10]. Additionally, proton-transfer-reaction mass spectrometry (PTR-MS) and flow infusion electrospray ionization mass spectrometry (FI-ESI-MS) are used to characterize the volatile and non-volatile components in nutmeg [17].

In Thailand, the nutmeg plant is primarily cultivated in specific areas of the country. The domestic consumption of nutmeg and mace in Thailand comprises their use in savory dishes, such as Thai massaman curry (Figure 1), and as herbal ingredients in various traditional medicinal formulas listed on the National List of Essential Medicines (NLEM). Our preliminary investigation showed two morphologic characteristics of the nutmeg sold in Thailand, which are either globose or oval shapes. Hence, this revelation heightens our awareness of the distinctions between plant types and the implications for consumer safety when they are ingested. However, the differences between these two types of nutmeg shapes, with respect to their DNA barcodes, phytochemical compositions, and toxicity evaluations, have not yet been investigated. Therefore, the aim of this study is to confirm whether these two nutmeg shapes differ using pharmacognostic analyses, and the results are confirmed via advanced chemical analysis. The identification of nutmeg types is conducted via DNA barcoding analysis, and the cytotoxic effects and the capacity of anti-nitric oxide production are compared.

## 2. Materials and Methods

The scheme of the methods used in this study is shown in Figure 2.

### 2.1. Sample Collections

Three authentic nutmeg samples (*Myristica fragrans* Houtt.) were obtained from different locations, including the medicinal plant garden of the Department of Medical Sciences, Ministry of Public Health, Chanthaburi Province, and nutmeg plantations in Chumphon and Songkhla Provinces (Table 1). Both arils and seeds were collected as reference samples, and branches containing important botanical structures, such as flowers and fruits, were assembled for herbarium specimen preparation. The authenticity of these *M. fragrans* samples was confirmed using established taxonomic information about *Myristica* species in Thailand [3], verified by Ms Wannaree Charaensup, a botanist of the Faculty of Pharmacy, Chiang Mai University (CMU). The herbarium specimens of the authentic *M. fragrans* (AL1-AL3) were maintained at the official herbarium of the Faculty of Pharmacy at CMU, with specimen numbers CMU0023365, CMU0023366, and CMU0023367. The harvested nutmeg arils and seeds were dried in a hot air oven at 50 °C until fully dehydrated (Appendix A). Additionally, for comparative analysis, commercial nutmeg samples (10 arils and 12 seeds) were procured from Thai spice markets to ascertain their botanical origins (Table 1; Appendix A).

### 2.2. Preparation of Nutmeg Extracts and Volatile Oils

Dried nutmeg arils and seeds were finely powdered using an electric blender. For methanolic extraction, 20 g of this powder was sonicated in 100 mL of methanol for 20 min and filtrated through Whatman No.4 filter paper. The filtrate was concentrated via rotary evaporation (EYELA, Tokyo, Japan). For the extraction of volatile oil, the hydro-distillation method outlined in THP 2022 [23] was employed. In this experiment, 10 g of nutmeg powder was placed in a 500 mL round-bottom flask with 200 mL of distilled water. The flask was connected to a distillation apparatus with a reflux condenser, and the mixture was heated using a heating mantle at 150 °C, distilling at approximately 3 mL per minute for 3 h. Subsequently, the mixture was left at a low temperature for over 1 h, leading to the separation of volatile oils, which were then prepared for subsequent analyses.

### 2.3. Pharmacognostic Identification of Nutmeg Samples

#### 2.3.1. Macroscopic Analysis

Both authentic and commercial nutmeg samples underwent macroscopic assessments, following the THP 2022 monograph [23]. Characteristics, including external appearances, aroma, and dimensions, were scrutinized. Measurements of aril length, width, and thickness, along with seed length and width, were taken using a Vernier caliper. These dimensional data were statistically analyzed via an unpaired *t*-test carried out in GraphPad Software (Prism 9) to differentiate both nutmeg groups.

#### 2.3.2. Thin-Layer Chromatography (TLC) Analysis

Methanolic extracts were prepared at a concentration of 1 mL in methanol, and a myristicin standard was similarly diluted in methanol. Each sample was applied as a 5 mm and 3 μL band using an automatic TLC sample (CAMAG^®^ Linomat 5, Muttenz, Switzerland). A precoated silica gel GF 254-activated aluminum plate (Merck, Dermstadt, Germany) was used during the stationary phase, and a mobile phase consisting of a 70:30 hexane–ethyl acetate mixture was employed [23]. The TLC procedure included development in a mobile phase saturated chamber over a 10 cm distance. After development, the TLC plate was dried and examined under UV light at 254 nm. Phytochemical constituents were visualized by spraying the plate with a 10% *v*/*v* sulfuric acid solution in absolute ethanol, followed by heating at 105 °C for 10 min. Final visualization and documentation were performed under UV 254 nm and white light.

### 2.4. Alternative Chemical Identification of Nutmeg

#### 2.4.1. High-Performance Liquid Chromatography (HPLC) Analysis

For a comparative analysis of the chemical fingerprint and myristicin quantities in the arils and seeds, HPLC was performed on two primary nutmeg shapes: globose and oval. Globose nutmeg was represented by samples AA1 (aril) and AS1 (seed), while oval-shaped nutmeg was represented by samples PA4 (aril) and PS3 (seed). The HPLC method, adapted from El-Alfy et al. [25], involved dissolving 20 mg of each extract and volatile oils in 1 mL of methanol, yielding concentrations between 2 and 20 mg/mL. After filtration through 0.45 μm membrane filters, a 10 μL aliquot was subjected to chromatography. Chemical fingerprint and myristicin quantities in the methanolic extracts and volatile oils were compared to a myristicin standard (Sigma-Aldrich, St. Louis, MO, USA) using an Agilent 1260 Infinity II System. This system included a quaternary pump, autoinjector, and PDA detector (Shimadzu, Tokyo, Japan). The myristicin standard was prepared at concentrations ranging from 12.5 to 400 μg/mL in 1 mL of methanol and filtered through a 45 μm membrane. Separation occurred on an Agilent ZORBAX Eclipse Plus C18 column (4.6 × 250 mm, 5 μM) (Agilent Technologies, Santa Clara, CA, USA) using a gradient elution over 32 min at a flow rate of 1 mL/min, with mobile phase A (100% acetonitrile) and B (0.1% aqueous formic acid). The elution gradient began with a 40:60 A:B ratio, transitioning to 80:20 by 2 min; and reaching 100:0 at 24 min. This was followed by a 5 min wash and re-equilibration, reverting to the initial 40:60 A:B ratio. HPLC data were captured at UV 280 nm using a photodiode array detector and processed in triplicate using the OpenLAB CDSChemStation Edition Software 3.5 (3.5.0) (Agilent Technologies, Santa Clara, CA, USA). Myristicin quantification was carried out based on comparisons with its standard curve.

#### 2.4.2. Gas Chromatography–Mass Spectrometry (GC-MS) Analysis

To elucidate the volatile chemical profiles of oils from the arils and seeds of both globose and oval-shaped nutmeg varieties, GC-MS analysis was conducted. Specifically, volatile oils from the aril (AA1) and seed (AS1) of the globose nutmeg, as well as the aril (PA4) and seed (PS3) of the oval-shaped variant, were subjected to scrutiny. Oil samples (500 ppm) were prepared in absolute methanol, filtered through 0.45 μm syringe filters, and transferred to 2 mL glass vial bottles. The separation of volatile oil components required the utilization of a QP2010 GC system (Shimadzu, Tokyo, Japan) with a 30 m × 0.25 mm i.d. × 0.25 μL film thickness column. Each sample was injected at a volume of 1 μL in the split sample mode (1:100), with helium as the carrier gas, at a flow rate of 1.23 mL/min. The column oven’s temperature was ramped from 60 to 210 °C at a rate of 3 °C/min. Mass spectral analyses used an electron ionization source under standardized conditions at 70 eV at 220 °C, with a quadrupole functioning as the ion analyzer. Compound identifications were derived from the U.S. National Institute of Standards and Technology (NIST) 14 Mass Spectra Library [26], supplemented by comparative analysis with established mass spectral data.

#### 2.4.3. Liquid Chromatography–Tandem Mass Spectrometry (LC-MS) Analysis

This analysis aimed to identify nonvolatile chemical constituents in the nutmeg extracts from both globose and oval-shaped groups. The LC-MS/MS method developed by Phosri et al. [27] was employed to analyze methanolic extracts from nutmeg samples, including the aril (AA1) and seed (AS1) from the globose nutmeg, as well as the aril (PA4) and seed (PS3) from the oval-shaped variant. Each extract, initially concentrated between 100 and 500 ppm, was diluted in absolute methanol and filtered through a 0.2 μm sterile filter. An Agilent 1290 UHPLC System (Agilent Technologies, USA) and an Agilent Poroshell EC-C18 (2.1 mm × 150 mm, 2.7 μm) column with Agilent Poroshell EC-C18 (4.6 mm × 5 mm, 2.7 μm) guard column was used for LC analyses. The column and autosampler temperatures were maintained at 30 and 5 °C, respectively. The injection volume was 1 mL, and the flow rate was set at 0.2 mL/min. The elution program employed a gradient of 0.1% (*v*/*v*) formic acid in deionized water (Phase A) and 0.1% (*v*/*v*) formic acid in acetonitrile (Phase B). The elution program was achieved as follows: t = 0 min, 5% B; t = 1 min, 5% B; t = 10 min, 17% B; t = 13 min, 17% B; t = 20 min, 100% B; t = 25 min, 100% B. For MS data acquisition, an Agilent G6454B Q-TOF Mass Spectrometer (Agilent Technologies, USA) was employed. The parameters were set as follows: gas temperature at 300 °C, gas flow at the rate of 11 l/min, nebulizer pressure at 45 psig, sheath gas temperature at 300 °C, and sheath gas flow at 12 L/min. The operation of the mass spectrometer relied on a Dual AJS ESI ion source, and the capillary (VCap) and nozzle voltages of the ion modes were set at 4000 V and 500 V, respectively. The voltages of skimmer1, fragmentor, and OctopoleRFPeak were maintained at 65 V, 150 V, and 750 V, respectively. The scan range covered was from 100 to 1100 *m*/*z* at a scan rate of 1.00 spectra/s. An Agilent reference mass solution consisting of specific internal reference compounds with an *m*/*z* of 121.05087300 and *m*/*z* of 922.00979800 for the positive mode and *m*/*z* of 112.98558700 and *m*/*z* 1033.98810900 for the negative mode was continuously infused in the MS using an Agilent 1260 isocratic pump. MS/MS acquisition further fine-tuned source parameters from the MS setup, adjusting the collision energy at 10, 20, or 40 eV. Compounds of MS/MS spectra were identified via matching with the METLIN database [28].

### 2.5. DNA Barcoding Analysis of Nutmeg Samples

#### 2.5.1. Genomic DNA Extraction

Authentic *M. fragrans* leaf samples (AL1–AL3) representing the globose nutmeg were finely pulverized using a mortar and pestle with the aid of liquid nitrogen, in preparation for genomic DNA extraction. In contrast, representative commercial nutmeg samples from the globose (PS1) and oval-shaped (PS3) groups were processed into a fine powder using an electric blender. Total DNA was then extracted from approximately 100 mg of each powder using the DNeasy Plant Mini Kit (Qiagen, Hilden, Germany) following the manufacturer’s protocol.

#### 2.5.2. Polymerase Chain Reaction Amplification and Sequencing

Polymerase chain reaction (PCR) was performed using a T960 PCR thermocycler (Drawell, Chongqing, China) to amplify five DNA barcoding regions: the internal transcribed spacer (ITS), maturase K (*mat*K), *trn*H-*psb*A intergenic spacer (*trn*H-*psb*A), ribulose-bisphosphate carboxylase (*rbc*L), and *trn*L-F regions. The PCR mixture included 100 to 200 ng of total DNA template, 2× PCR Buffer for KOD FX Neo, nuclease-free water, KOD Fx Neo polymerase (TOYOBO Life Science, Osaka, Japan), and both forward and reverse primers specific to each DNA barcoding region (the details are in Appendix A). Specific PCR conditions are provided in the Appendix A. Products were verified by analyzing them using 1.8% *w*/*v* agarose gel, staining with RedSafe™ nucleic acid staining solution (iNtRON Biotechnology, Seongnam, Republic of Korea), and visualizing under UV light using the Gel Documentation EZ Imager (Bio-Rad, Hercules, CA, USA). Subsequently, PCR products were purified and subjected to bidirectional sequencing on an ABI PRISM 3730 XL sequencer (Applied Biosystems, Waltham, MA, USA).

#### 2.5.3. Bioinformatics Analysis

Nucleotide sequences were manually curated using BioEdit Version 7.2.5 [29] and MEGA Software Version 11.0.10 [30]. They were aligned with MUSCLE Software 3.8.31 to pinpoint the most suitable DNA barcoding region for commercial nutmeg identification. The refined sequences were then cross-referenced with the GenBank database via the Basic Local Alignment Search Tool (BLAST) [31]. As of 20 July 2023, sequences with a minimum of 97% identity were considered top matches [32]. For phylogenetic analysis, the neighbor-joining (NJ) method supported by 10,000 bootstrap replications with the K2P distance model was employed to assess tree topology.

### 2.6. Cytotoxic Assessment of Nutmeg (M. fragrans) and Its Substituent (M. argentea)

#### 2.6.1. Preparation of Methanolic Extracts and Volatile Oils from Nutmeg (*M. fragrans*) and Its Substituent (*M. argentea*)

Cytotoxic tests were conducted on representative methanolic extracts and volatile oils from nutmeg samples, including the aril (AA1) and seed (AS1) of *M. fragrans* and the aril (PA4) and seed (PS3) of *M. argentea*. Sample preparation followed Champasuri and Intharat’s protocol with slight modifications [33]. The extracts were dissolved in sterile dimethyl sulfoxide (DMSO) and diluted to concentrations ranging from 12.5 to 200 μg/mL.

#### 2.6.2. Cell Culture

Mouse embryonic fibroblast cells (3T3-L1), human intestinal epithelial cells (Caco-2), human embryonic kidneys 293 cells (HEK93), macrophage cells (RAW264.7), and human keratinocyte cells (HaCaT) were obtained from the American Type Culture Collection (ATCC, Manassas, VA, USA). Cells were cultured at 37 °C, 5% CO_2_, and 95% humidity in Dulbecco’s Modified Eagle Medium (DMEM, Sigma, St. Louis, MO, USA). This medium was supplemented with 10% fetal bovine serum (FBS, Sigma, St. Louis, MO, USA) and a combination of penicillin (100 U/mL) and streptomycin (100 μg/mL) from Gibco, Thermo Fisher Scientific, Grand Island, NY, USA. While most cells were propagated in cell culture flasks, RAW264.7 cells were cultured in ultra-low attachment culture dishes.

#### 2.6.3. Determination of Cell Viability Using the Sulforhodamine B (SRB) Assay Protocol

Cytotoxicity was assessed in various cell lines to evaluate the effects of *M. fragrans* and *M. argentea* on both topical and internal human systems. The assessment followed Prakash and Gupta’s protocol with minor modifications [34]. Cell lines, including 3T3-L1 (3 × 10^3^ cells/well), Caco-2 (5 × 10^3^ cells/well), HEK293 (3 × 10^3^ cells/well), RAW264.7 (1.5 × 10^3^ cells/well), and HaCaT (3 × 10^3^ cells/well) were seeded. Each was incubated with 100 μL of extract for 24 (HaCaT and RAW264.7) and 48 (3T3-L1, Caco-2, HEK293, and RAW264.7) hours. After incubation, 20% *w*/*v* trichloroacetic acid (TCA) at 100 μL was added to each well, and the samples were chilled at 4 °C for 1 h. Following TCA removal via washing, 100 μL of 0.057%SRB was added. After five washes with 200 μL of 1% acetic acid to remove excess SRB, the SRB–protein complex was dissolved in 200 μL of Tris-buffer. Absorbance was read at 570 nm using a microplate reader (BioTek Synergy H4, Winooski, VT, USA). Cell viability was calculated as %cell viability = 100 × (OD sample with SRB/OD control media with SRB), and IC_20_ and IC_50_ values were determined.

### 2.7. Nitric Oxide (NO) Assay

To compare the property of NO production inhibition between *M. fragrans* and *M. argentea*, RAW 264.7 cells were seeded and cultured in 96-well plates for 24 h. After this period, the culture medium was replaced with fresh medium, and the cells were stimulated with 10 ng/mL of LPS to induce inflammation. Subsequently, cells were treated with 100 μL of each extract at a non-toxic dose for an additional 24 h. NO production was assessed by combining Griess reagents with the culture supernatant, and absorbance was read at 540 nm using a microplate reader. The percentage of NO production was calculated as %NO production = 100 × (NO release with sample − spontaneous release)/(NO release without sample − spontaneous release).

### 2.8. Statistical Analysis

To determine differences between the treatment categories on toxicity and anti-nitric oxide production tests, a Student’s *t*-test was employed for dual groups, while one-way ANOVA was applied for groups of three or more. Results are shown as average ± SEM. For all evaluations, significance was recognized at a *p*-value of less than 0.05. All statistical analyses were facilitated via GraphPad software version Prism 9 (GraphPad Software, Boston, MA, USA)

## 3. Results

### 3.1. Pharmacognostic Authentication of Nutmeg Samples

#### 3.1.1. Macroscopic Examination

Both authentic *M. fragrans* and commercial nutmeg samples, including arils and seeds, were assessed for their macroscopic characteristics concerning their external appearance, odor, and size, in line with guideline THP2022 [23]. Despite variances, all nutmeg samples showed similar external appearances and odors. Specifically, the arils manifested as yellowish-brown to brownish, flattened, and fragmented sections that emanate a pungent scent. Conversely, the seeds ranged from oval to globose forms, presenting a brownish hue accompanied by a distinct pungent aroma (refer to Figure 3 and Appendix A). Upon carrying out measurements, the arils were segregated in two distinct categories based on size and form. The first category comprised globose arils with average dimensions of 2.79 × 0.97 × 0.047 cm. The second category included oval-shaped arils, with average dimensions of 3.80 × 1.02 × 0.056 cm. Similarly, the seeds were split into two discernible groups based on form: globose seeds (averaging 2.03 × 1.74 cm) and oval-shaped seeds (averaging 3.04 × 1.65 cm). Notably, the shape and size of the authentic *M. fragrans* arils and seeds aligned with the globose category. Given the discernible macroscopic variations between the two categories, the results revealed that while the width remained consistent across the two groups, significant differences emerged concerning their lengths (*p*-value < 0.05).

#### 3.1.2. Thin-Layer Chromatography Analysis

To further delineate the differentiation of nutmeg into two distinct groups, a comprehensive chemical assessment using TLC was employed. Distinct chromatographic patterns emerged when analyzing the TLC chromatograms of both aril and seed extracts. As illustrated in Figure 4a–d, these TLC chromatograms, observed under UV 254 nm and white light post spraying with 10% sulfuric acid in absolute ethanol, presented clear differences. Notably, all methanolic aril extracts matched the myristicin standard with an hRf of 80. Additionally, this chemical analysis presented two different chromatographic patterns found in both aril and seed extracts: one pattern was the same with respect to the extract of authentic samples, and another was not the same. The chromatographic TLC patterns of the methanolic extracts of globose arils (PA1 to PA3 and PA6) and globose seeds (PS1 to PS2, PS4, PS6, PS8, and PS11) were the same as those of the authentic extracts (AA1 to AA3 and AS1 to AS3), while the extracts of oval-shaped arils (PA4 to PA5 and PA7 to PA10) and oval-shaped seeds (PS3, PS5, PS7, PS9-PS10, and PS12) showed different patterns compared to those of the extracts of authentic samples, and also presented remarkable spots at hRf = 65 (black arrow), which could be found in both analyses (Figure 4a–d). Therefore, the TLC results align with the morphologic findings, reinforcing the categorization of the arils in two groups based on their size and shape.

### 3.2. Alternative Chemical Identification of Nutmeg Samples

In order to confirm the analytical results of the pharmacognostic method, advanced chemical analysis methods were employed to analyze the differences in the chemical compounds found in both types of nutmeg. These methods include HPLC for the analysis of chromatographic fingerprints and key compounds such as myristicin; GC-MS for the analysis of chemicals found in the volatile oil; and GC-MS/MS for the analysis of compounds found in ethanolic extracts.

#### 3.2.1. High-Performance Liquid Chromatography Analysis

The HPLC technique was employed to discern the phytochemical profiles of the methanolic extracts (Figure 5a–d) and volatile oils (Figure 5e–h) of the representative nutmeg seed and aril samples, and they were categorized with respect to their globose and oval shapes. The focal point of this analysis was myristicin, a principal compound in nutmeg. The calibration curve for the exhibited robust linearity ranged between concentrations of 12.5 and 400 μg/mL. The derived linear regression equation is y = 0.0028x − 0.0199, where x represents myristicin concentrations (μg/mL) and y corresponds to the peak area. The determination coefficient (R^2^) was 0.9999. The HPLC chromatogram (Figure 4a to h) pinpointed the myristicin peak at a retention time of 17.35 min. In evaluating the myristicin contents of these samples, the seed extract from globose nutmeg (AS1) exhibited the highest concentration: 25.54 ± 2.37 mg/g extract. This was followed by the seed extract of the oval-shaped nutmeg (PS3) at 19.35 ± 1.44 mg/g extract. For arils, the globose nutmeg (AA1) contained 10.60 ± 0.35 mg/g extract, while its oval-shaped counterpart (PA4) contained 3.76 ± 0.52 mg/g extract. Similarly, volatile oil evaluations indicated the highest myristicin content in the seeds of globose nutmeg (AS1) (53.42 ± 2.10 mg/g extract), followed by the arils of globose nutmeg (AA1) (40.48 ± 1.58 mg/g extract), arils of oval-shaped nutmeg (PA4) (5.35 ± 0.18 mg/g extract), and seeds of oval-shaped nutmeg (PS3) (4.74 ± 0.14 mg/g extract). Further examination of HPLC chromatographic patterns revealed distinct peaks that could differentiate the globose nutmeg from oval-shaped nutmeg. As depicted in Figure 5a–h, a noticeable peak at a retention time of 18.30 min was prevalent in both the arils and seeds of the oval-shaped nutmeg. In addition, a pronounced peak at a retention time of 11.00 min was evident in volatile oils from oval-shaped nutmeg samples (Figure 5g,h), whereas lower peaks were observed in globose nutmeg samples at the same retention time (Figure 5e,f). This differential HPLC fingerprinting operation substantiates the feasibility of distinguishing between globose and oval-shaped nutmeg samples.

#### 3.2.2. Gas Chromatography–Mass Spectrometry Analysis

The volatile oils from the seed and aril of both globose and oval-shaped nutmegs were analyzed using the GC-MS technique. Detailed GC chromatograms can be found in Appendix A. The area under the curve percentage (%AUC) of major peaks within each chromatogram was identified via cross-referencing with the NIST 14 mass spectra library. Notably, dominant peaks were associated with pinene, β-phellandrene, and terpinen-4-ol, which appeared at retention times of 6.7, 7.9, and 15.7, respectively. Two chemical marker peaks were identified as safrole and myristicin, registering at retention times of 20.4 and 30.1, respectively. Focusing on the area under the curve (AUC) percentages of myristicin and safrole, distinct differences emerged between globose and oval-shaped nutmegs. Specifically, both the aril (AA1) and seed (AS1) of globose nutmegs demonstrated a high AUC percentage for myristicin but a low AUC percentage for safrole (Table 2). On the contrary, both the aril (PA4) and seed (PS3) of oval-shaped nutmegs showed the opposite pattern, with a heightened AUC for safrole but a diminished AUC for myristicin. Statistical analysis revealed that only the AUC percentage of myristicin in the aril significantly distinguished globose nutmegs from their oval-shaped counterparts (*p* < 0.05) (Table 2).

#### 3.2.3. Liquid Chromatography–Tandem Mass Spectrometry (LC-MS) Analysis

The LC-MS/MS method was employed to discern and compare the nonvolatile phytochemical constituents present in the methanolic extracts of arils and seeds from both globose and oval-shaped nutmeg samples. The key compounds detected in these extracts are summarized in Table 3. In comparing the aril extracts of both nutmeg shapes, nine primary compounds were identified. Two compounds, malabaricone C and dimorphecolic acid, were exclusive to the globose nutmeg (AA1) aril extract. In contrast, β-D-glucose, N-((5-(dimethylamino)-1-naphthyl)sulphonyl)-L-histidine, sucrose, phosphatidylinositols, and 1-oleoylglycerophosphoinositol were uniquely present in the aril extract of the oval-shaped nutmeg (PA4). Regarding seed extracts, twelve principal compounds were discerned. Four of these, C1’-C9-Glycosylated UWM6, 8-desoxygartanin, clivorine, and cheirotoxol, were distinct to the globose nutmeg (AS1) seed extract. However, phospholipid derivatives, specifically phosphatidylserine and phosphatidic acid, were detected solely in the seed extracts of oval-shaped nutmeg samples (PS3).

### 3.3. DNA Barcoding Analysis of Nutmeg Samples

To ascertain the botanical origins of the two different nutmeg samples, DNA barcoding was employed. Authentic *M. fragrans* leaf identification was carried out using DNA barcoding analysis. Five DNA barcoding regions—ITS, *mat*K, *rbc*L, *trn*H-*psb*A and *trn*L-F—were considered potential markers for distinguishing closely related *Myristica* species. The authentic *M. fragrans* samples (AL1 to AL3) were amplified for all regions, producing clear nucleotide sequences across all but the ITS region. These sequences were then cross-referenced with the reference nucleotide sequence database on NBCI’s GenBank via the BLAST interface. The results, detailed in Table 4, indicated large similarities between the sequence of authentic *M. fragrans* and the GenBank database references. While *mat*K, *rbc*L, and *trn*L-F regions all showed a 100% identity match with both *M. fragrans* and other *Myristica* species, the *trn*H-*psb*A regions were particularly revealing. Here, the highest match was with *M. fragrans* at 99.44 and 98.89% identity, followed by *M. yunnanensis* and *M. teysmannii* with 97.49% and 96.21% identity, respectively. This demonstrated that the *trn*H-*psb*A regions exhibited the greatest variance among the five DNA barcode regions. Hence, *trn*H-*psb*A was identified as the most suitable DNA barcoding region for distinguishing between globose and oval-shaped nutmeg samples.

The *trn*H-*psb*A region’s nucleotide sequences of representative nutmeg samples from the globose (PS1) and oval-shaped (PS3) groups were analyzed. A sequence alignment of this region for authentic *M. fragrans* and both representatives is presented in Appendix A. These sequences underwent BLAST and neighbor-joining tree topology analyses to ascertain the botanical origins of the samples. Using the BLAST tool against the GenBank database, the globose nutmeg (PS1) exhibited 99.44% BLAST homology with respect to *M. fragrans*. In contrast, the oval-shaped nutmeg (PS3) presented 100% BLAST homology with respect to *M. argentea* (OP866724). These results are detailed in Table 2. Further analyses using *trn*H-*psb*A sequence data led to the construction of a neighbor-joining tree, incorporating sequences from authentic *M. fragrans*, the globose nutmeg, the oval-shaped nutmeg, and various *Myristica* species retrieved from the GenBank database (Figure 6). This dendrogram revealed that the globose nutmeg (PS1) was clustered with *M. fragrans*, while the oval-shaped nutmeg (PS3) aligned closely with *M. argentea*. Consequently, it was deduced that the globose nutmeg originates from *M. fragrans*, while the oval-shaped nutmeg is linked to *M. argentea*. These findings suggest that the *trn*H-*psb*A region is instrumental in distinguishing *M. fragrans* from *M. argentea*, and that it can be used to authenticate the botanical origins of commercial nutmeg.

### 3.4. Cytotoxicity Tests of the Arils and Seeds of Nutmeg and Its Substituents

In assessing the cytotoxic properties of various nutmeg samples, our analysis encompassed both methanolic extracts and volatile oils, with an aim to understand their impact on a variety of cell lines. Cytotoxicity was quantified via IC_20_ and IC_50_ values. Different cell lines such as 3T3-L1, Caco-2, HEK 293, and RAW264.7 cells were used to represent internal organs: fibroblasts, gastrointestinal tracts, kidneys, and the immune system, respectively (Figure 7a–h). In contrast, HaCaT cells signified an external organ: the skin (Figure 7i,j). Notably, HaCaT cytotoxicity highlighted the volatile oils that maintained above 90% cell viability even at doses as high as 150 μg/mL, resulting in the HaCaT cell calculation of IC20 and IC50 for this oil (Figure 7j and Table 5).

Drawing comparisons between similar species’ extracts but employing different extraction methods (methanolic extracts vs. volatile oils), the data elucidated that volatile oils generally presented a lower IC_20_ than their methanolic counterparts. Specifically, when comparing the aril methanolic extracts of *M. fragrans* (AA1) and *M. argentea* (PA4), *M. fragrans* consistently exhibited a reduced IC_20_ across all cell lines. A similar trend was discernible with seed methanolic extracts of *M. fragrans* (AS1) and *M. argentea* (PS3), although an exception was observed in HaCaT cells, where AS1 manifested a higher IC_20_ than PS3.

Considering the comparative toxicity of methanolic extracts and volatile oils across all cells, methanolic extracts typically exhibited greater toxicity. AA1′s methanolic extract showed the lowest IC_20_ in 3T3-L1 (25.3 μg/mL), HEK 293 (20.1 μg/mL), Caco-2 (10.3 μg/mL), and RAW264.7 (8.2 μg/mL) cells (Table 6). In contrast, the methanolic extracts of PS3 manifested the highest toxicity in HaCaT with an IC_20_ = 6.6 μg/mL (Table 5).

Before carrying out assessments using the NO assay, the cytotoxicity of both methanolic extracts and volatile oils on RAW 264.7 cells was determined. The cells were exposed to a concentration range from 12.5 to 200 μg/mL for 24 h, followed by an SRB assay. Notably, cell viability showed a dose-dependent increase, with methanolic extracts exhibiting pronounced toxicity on RAW264.7. Based on these cytotoxicity results, suitable concentrations that ensured a cell viability of ≥80% or above (IC_20_) were established: methanolic crude extract AA1 was set at 12.5 μg/mL; AS1, PA4, and PS3 were at 25.0 μg/mL; all volatile oils were at 100.0 μg/mL; and the standard myristicin compound was set at 200.0 μM for the NO production inhibition test.

### 3.5. Results from the Nitric Oxide (NO) Production Inhibition Test

In the NO production inhibition test, we used established non-toxic concentrations for each sample extract. Figure 8 presents the results of this test with respect to RAW264.7 cells that exhibited inflammation induced by LPS. Notably, the methanolic extracts of AA1 and AS1 showed statistically significant inhibitions (*p* < 0.05) compared with the control group or LPS alone. The methanolic extract derived from nutmeg arils, specifically AA1 from *M. fragrans* with a globose shape, exhibited the highest NO production inhibition in RAW 264.7 cells (37.26%). This was followed by the methanolic extract of the oval-shaped aril (PA4) at 29.52%. Interestingly, the myristicin standard did not yield statistically significant results.

## 4. Discussion

### 4.1. Integrative Approaches Offer Precision in Discriminating M. fragrans from Its Substituent

Based on our initial investigations, it has come to our attention that the nutmeg available on the market exhibits two distinct characteristics. Consequently, the authentication of both types of nutmeg has become imperative. The Thai Herbal Pharmacopoeia 2022 recommends both macroscopic and microscopic techniques for the comprehensive morphological assessment of both nutmeg arils and seeds [23]. Our macroscopic investigations simply identified two distinct morphologies in the nutmeg available in Thai spice markets: globose and oval-shaped nutmeg. However, when dealing with powdered nutmeg products, traditional morphologic identification proves inadequate and the experience of an inspector is required. Evidence has shown that an integrative approach can bridge the shortcomings of a singular identification method [35,36]. Consistent with this idea, our research employed TLC to obtain a nuanced understanding of the phytochemical constituents of mace and nutmeg. In this study, TLC fingerprints showed the different patterns between globose and oval-shaped nutmeg samples. These TLC chromatographic distinctions mirrored the macroscopic variations, validating the two primary groupings observed in our samples. This synergy between morphologic and TLC chromatographic data underlines the value of using a multi-faceted approach in botanical identification. These methods are simple and can be used for the preliminary identification of nutmeg products. Nonetheless, it is essential to acknowledge that environmental variability can impact the concentrations of key compounds in plants. Therefore, employing advanced techniques will enhance the robustness of our study’s findings.

The application of advanced chemical analytical methods, specifically HPLC, GC-MS, and LC-MS/MS, provided conclusive data for distinguishing *M. fragrans* from its potential adulterants. HPLC, a widely adopted tool, is recognized for its efficacy in ensuring the quality of various food products, spices, and herbal medicines [37]. Our investigations highlighted that both globose and oval-shaped nutmeg samples were characterized by the presence of myristicin, a primary phytochemical marker of *M. fragrans* [38]. Distinctive chromatographic patterns observed in the HPLC fingerprints further corroborated our ability to differentiate between the two nutmeg groups. This result is consistent with related reports that use HPLC fingerprints to ascertain the purity of spices like paprika powder [39], saffron [40], and cassia bark [41]. GC-MS chromatographic fingerprints provide a detailed profile of the volatile chemical constituents in samples [42]. In this study, the chromatographic pattern obtained via GC-MS/MS revealed differences in chemical marker peaks, particularly safrole and myristicin. A high area under the curve percentage (%AUC) for myristicin and a low %AUC for safrole were observed in both the aril (AA1) and seed (AS1) of globose nutmeg samples. Conversely, a high %AUC of safrole and a low %AUC of myristicin were detected in both the aril (PA4) and seed (PS3) of the oval-shaped nutmeg. This finding is consistent with Oyen and Nguyen’s findings, in which *M. fragrans* was characterized by high myristicin and low safrole contents, while *M. argentea* exhibited high safrole and low myristicin contents [43]. Recently, LC-MS/MS, an advanced analytical method, has been crucial in verifying the authenticity of spices, as demonstrated in oregano [44], saffron [45], and chili pepper [46]. Our analysis discerned unique nonvolatile compounds within globose and oval-shaped nutmeg samples. For instance, globose nutmeg aril exclusively contained malabaricone C and dimorphecolic acid, while compounds like clivorine and cheirotoxol were only detected in its seed. These unique chemical markers can serve as definitive discriminants between globose and oval-shaped nutmeg variants. However, advanced chemical analysis (HPLC, GC-MS, and LC-MS/MS) was able to accurately determine the two types of nutmegs. These analytical methods require specific solvents and sophisticated instruments, rendering them costly. Additionally, chemical analysis methods may not accurately identify plant species. Therefore, DNA barcoding was employed to establish the correct identification of both nutmeg species.

DNA barcoding, employing short genomic DNA fragments for species identification, signifies a modern advancement in biological identification techniques [22]. It offers a marked enhancement over traditional methods, substantiating their findings and improving precision [47]. The application of DNA barcoding in ensuring the authenticity of spices and food products has been well documented, with several studies underscoring its efficacy in distinguishing genuine nutmeg from potential adulterants [20,48]. In this study, we explored five primary DNA barcoding regions-ITS, *mat*K, *rbc*L, *trn*H-*psb*A, and *trn*L-F, with the goal of authenticating the commercial nutmeg samples available in Thailand. Interestingly, we encountered challenges in sequencing the nucleotide sequence of *M. fragrans* using the ITS region, a finding that aligns with the observations reported in Swetha et al.’s study [20]. Via BLAST analysis, the *trn*H-*psb*A region emerged as the most promising DNA barcoding segment for nutmeg authentication because it allows for the differentiation of closely related *Myristica* species, which distinguishes it from other regions. Subsequent analyses of representative samples, the globose seed (PS1) and the oval-shaped seed (PS3) were analyzed using the DNA barcoding of the *trn*H-*psb*A region. According to the BLAST results and the neighbor-joining tree analysis, the results confirmed the botanical origins of PS1 as *M. fragrans* and PS3 as *M. argentea*. Hence, it is evident that BLAST and phylogenetic tree methods serve as valuable tools for the identification of various plant species, including, but not limited to, *Mucuna* spp. [37], *Cassia* spp. [49], and *Sida cordifolia* [50]. According to these findings, DNA barcoding constitutes a powerful tool for identifying and authenticating nutmeg varieties. This discovery will exert an influence on forthcoming research endeavors aimed at devising prompt, replicable, and precise tools. Examples of such tools include immunochromatographic assays or DNA-chromatographic detection strips, which can be instrumental in enhancing food safety applications.

### 4.2. Comparing Cytotoxicity Tests on M. fragrans and Its Substituent (M. argentea)

In the Thai spice market, there is another species of nutmeg (*M. argentea*) that differs from *M. fragrans*; this observation has raised awareness regarding the safety of consuming *M. argentea* nutmeg. Therefore, researchers are interested in conducting a comparative study on the toxicity and biological activity of both nutmeg types. Cytotoxicity tests revealed that volatile oils, relative to methanolic extracts, exhibited lower toxicity. This finding aligns with the US Food and Drug Administration’s (FDA) classification with respect to how safe nutmeg volatile oil substances are for consumption [51]. Specifically, volatile oil extracts showed negligible toxicity relative to HaCaT cells, reflecting their common applications in products like cosmetics, perfumes, or toiletries [52]. As confirmed by 24 hr cytotoxic tests on HaCaT cells, aligned with related studies [53,54], the potential of these oils for proliferation or wound healing remains a topic for future investigation.

Focusing on the methanolic extracts, those derived from *M. fragrans* (AA1 and AS2) showed lower IC_20_ values compared to *M. argentea* (PA4 and PS3) in cells representing internal organs, including 3T3-L1, Caco-2, HEK 293, and RAW 264.7. In contrast, concerning HaCaT cells, the IC_20_ values were higher for AA1 and AS1 extracts. The existing scientific literature indicated that the ethanolic extract of *M. fragrans* can induce the expression of uncoupling proteins (UPC1 and UPC2). These proteins, in turn, modulate the peroxisome proliferator-activated receptor γ (PPARγ), which may influence the proliferation and characteristics of white adipose tissue in 3T3-L1 cell lines [55]. Furthermore, both AA1 and AS1 extracts exhibited a trend of lower IC_20_ values in Caco-2. Notably, these extracts, exhibiting a higher myristicin content, showed lower IC_20_ values than PA4 and PS3 extracts with lower myristicin contents. This observation aligns with studies that suggest the anti-proliferative properties of nutmeg’s fixed oil are obtained via supercritical fluid extraction and its active compound, myristicin, especially with respect to Caco-2 cells [56,57].

In examining NO production inhibition, methanolic extracts from AA1, PA4, and AS showed reduced NO production compared with that of standard myristicin, which is known for its anti-inflammatory properties [57]. Interestingly, while AS1 exhibited the highest myristicin contents among the extracts, it suggests that myristicin might not be the primary bioactive compound responsible for the anti-inflammatory properties in nutmeg. On another note, safrole has been linked to an increase in radical superoxide generation, potentially inducing inflammation in RAW264.7 cells. However, despite its higher safrole content, the methanolic extract from PA4 was observed to decrease NO production in RAW 264.7. This could be attributed to the interaction between safrole and other compounds present in PA4 [58]. To further pinpoint the primary active compound in nutmeg with anti-inflammatory properties, it would be beneficial to test isolated substances derived from fractionate extracts.

## 5. Conclusions

The nutmeg available in Thailand is sourced from two distinct morphological shapes: globose and oval configurations. These can be distinguished using pharmacognostic methods such as macroscopic and TLC chromatographic methods. Furthermore, advanced chemical analyses, namely HPLC, GC-MS, and LC-MS/MS confirmed these results. Based on the BLAST results and neighbor-joining tree analysis, the DNA barcoding of the *trn*H-*psb*A region identified the botanical origin of these two distinct nutmegs: *M. fragrans* and *M. argentea*. Thus, we raise concerns regarding its therapeutic properties and toxicity, which influence the confidence of consumers. However, both methanolic extracts from the arils of *M. fragrans* and *M. argentea* exhibit anti-inflammatory properties and have similar cytotoxicity profiles. The substitution or blending of these two species appears to be safe and does not compromise their therapeutic potential. These findings instill confidence in consumers regarding the safe use of these plant species. Nonetheless, within the perfume industry, the discernible differences in the chemical compositions of these two plants may exert an influence on the overall perfume quality. Consequently, the development of future tools characterized by simplicity, speed, and a high degree of accuracy, namely immunochromatographic assays or DNA-chromatographic detection strips that distinguish between these two plant types, holds significant importance within the perfume industry.

## Figures and Tables

**Figure 1 foods-12-04211-f001:**
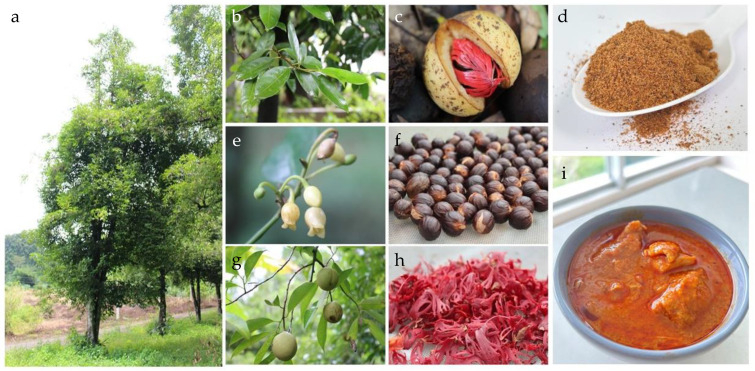
Morphologic characteristics of *M. fragrans* Houtt.: (**a**) habit, (**b**) leaves, (**c**) ripened fruit displaying the aril, (**d**) ground nutmeg, (**e**) flowers, (**f**) seeds, (**g**) fruits, (**h**) dried arils, and (**i**) Thai massaman curry.

**Figure 2 foods-12-04211-f002:**
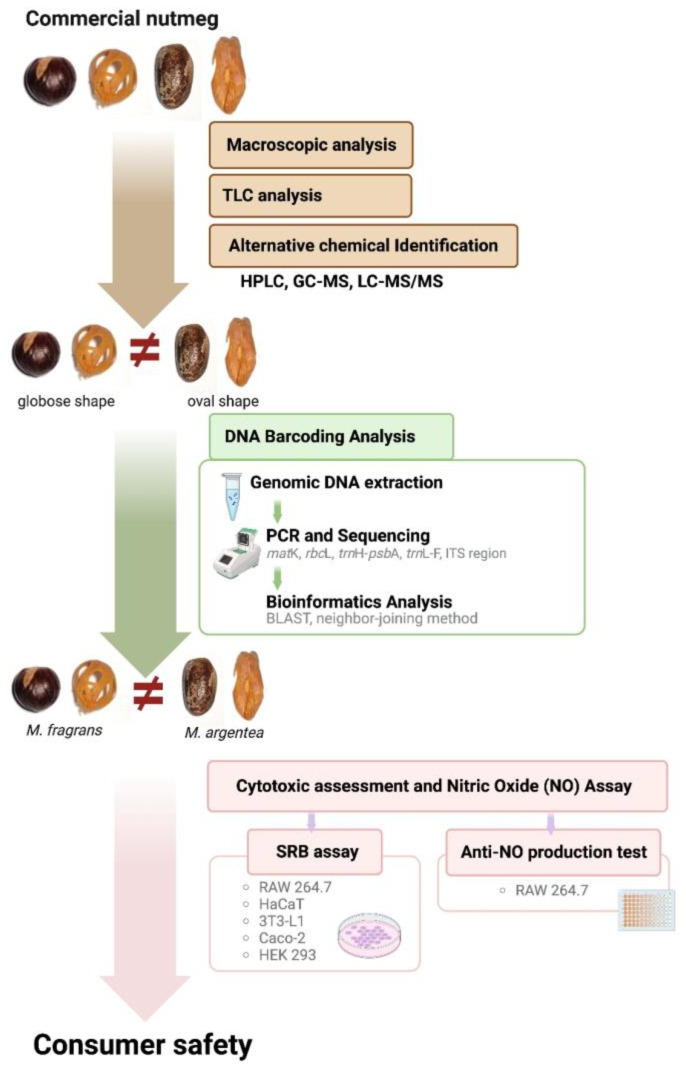
The methodological framework employed in this study.

**Figure 3 foods-12-04211-f003:**
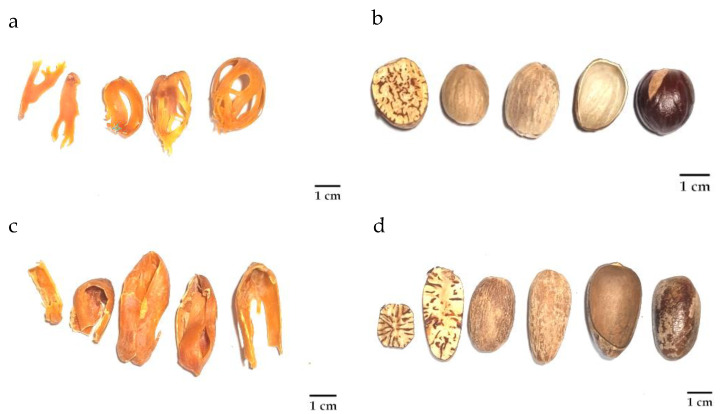
Macroscopic features of nutmeg arils and seeds divided in two groups: globose (**a**,**b**) and oval-shaped groups (**c**,**d**).

**Figure 4 foods-12-04211-f004:**
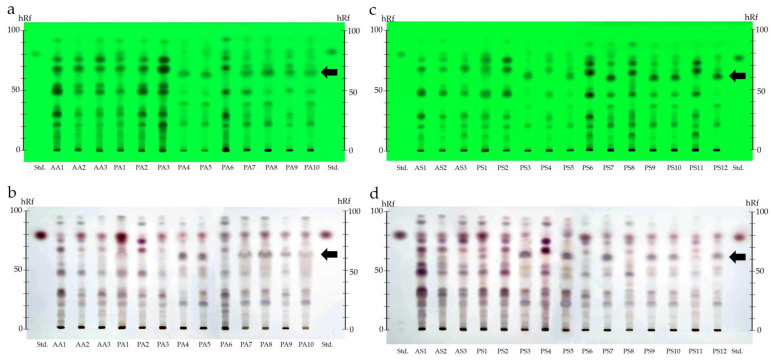
TLC fingerprints of authentic *M. fragrans* arils (AA1–AA3) and seeds (AS1–AS3), along with commercial nutmeg arils (PA1–PA10) and seeds (PS1–PS12), were developed using hexane: ethyl acetate (70:30) as the mobile phase. All samples were compared using the standard myristicin (Std.). The TLC chromatograms of aril and seed samples, detected under UV at 254 nm ((**a**) for arils and (**c**) for seeds) and after being derivatized with 10% *v*/*v* sulfuric acid ((**b**) for arils and (**d**) for seeds), are presented. Black arrows indicate the presence of bands at hRf = 65 in all oval-shaped samples.

**Figure 5 foods-12-04211-f005:**
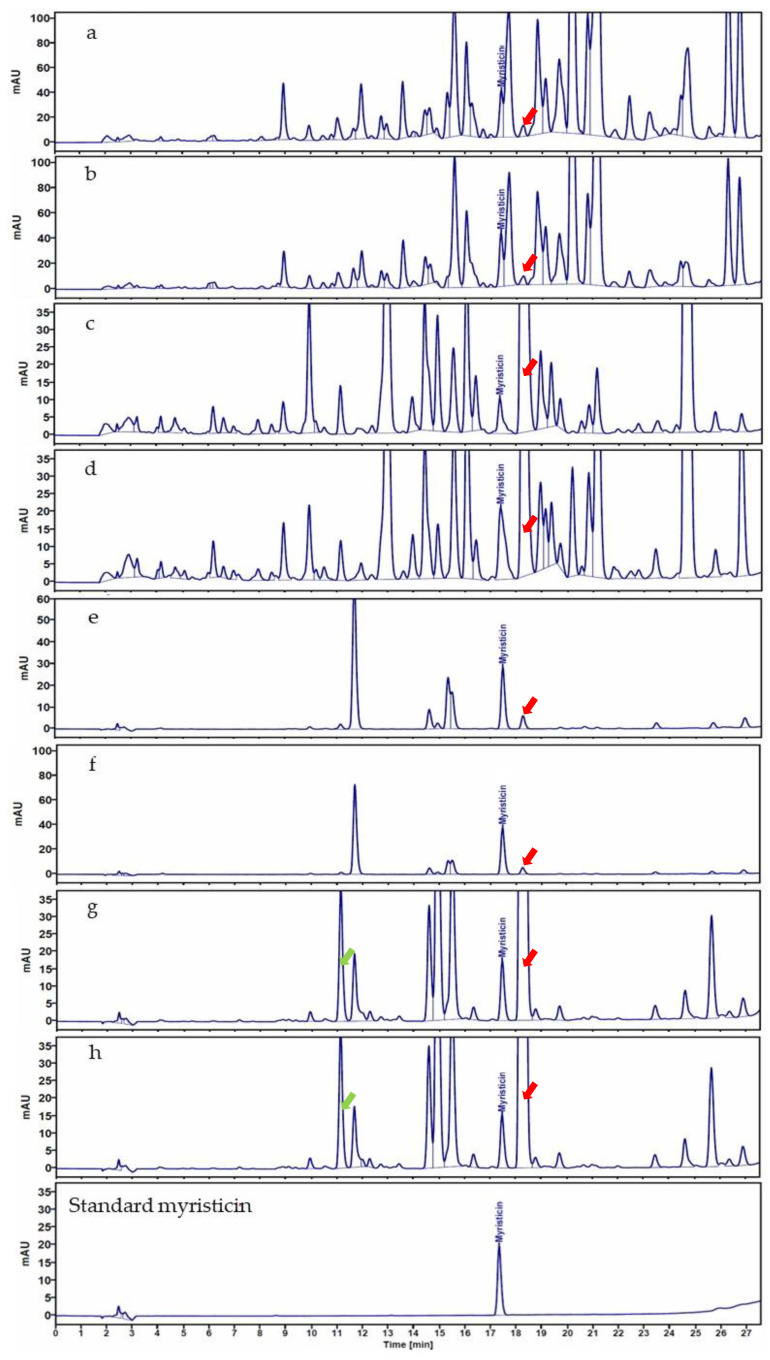
HPLC fingerprints of the methanolic extracts (**a**–**d**) and volatile oils (**e**–**h**) from representative nutmeg samples of a globose (**a**,**b**,**e**,**f**) and an oval shape (**c**,**d**,**g**,**h**). The green and red arrows indicate distinct peaks at retention times of 11.00 and 18.30 min, respectively, distinguishing the globose nutmeg samples from the oval-shaped ones.

**Figure 6 foods-12-04211-f006:**
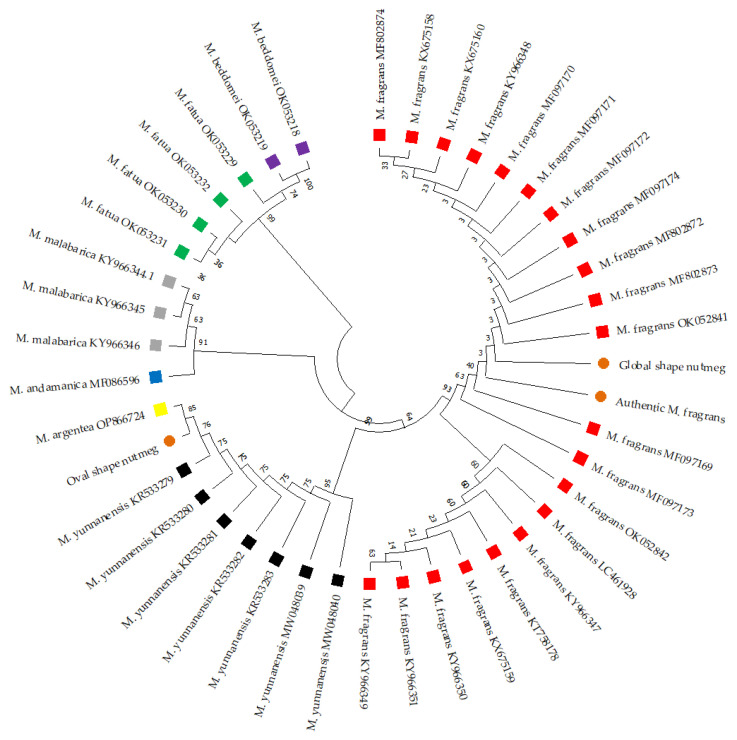
Neighbor-joining tree constructed using MEGA 11.0.10 Software, showcasing authentic *M. fragrans*, globose nutmeg (PS1), and oval-shaped nutmeg (PS3), alongside various *Myristica* spp. retrieved from the GenBank database. The *Myristica* species from the GenBank database include *M. fragrans* (highlighted in red), *M. yunnanensis* (black), *M. argentea* (yellow), *M. andamanica* (blue), *M. malabarica* (grey), *M. fatua* (green), and *M. beddomei* (violet). The orange dots represent the identified *Myristica* specimens from our study. The dendrogram was constructed based on the aligned nucleotide sequences from the *trn*H-*psb*A region. Numerical values at the nodes represent bootstrap values derived from 10,000 replications.

**Figure 7 foods-12-04211-f007:**
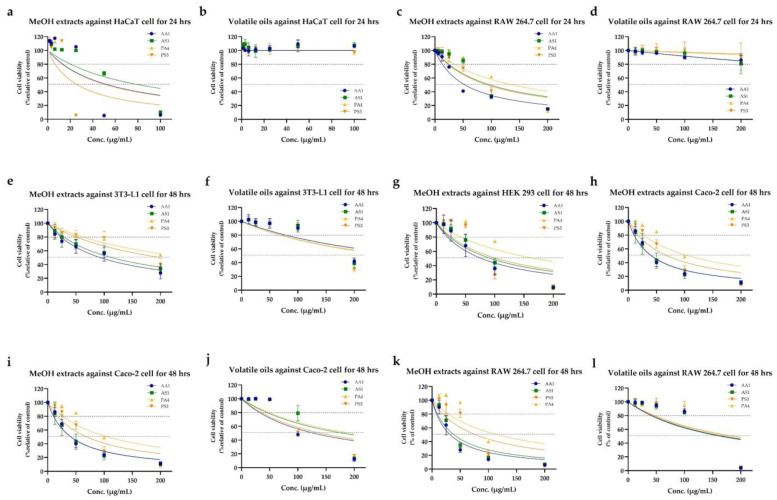
Assessment of the cytotoxic effects of methanolic extracts and volatile oils extracted from *M. fragrans* and its substituent, *M. argentea*: representations include *M. fragrans* aril (AA1; blue) and seed (AS1; green), along with *M. argentea* aril (PA4; orange) and seed (PS3; red). After 24 h treatment, the SRB assay illustrates the effect of the methanolic extracts and volatile oils of *M. fragrans* and *M. argentea* on the viability of HaCaT (**a**,**b**) and RAW 264.7 (**c**,**d**) cells, and illustrates the effect after 48 h treatment on 3T3-L1 (**e**,**f**), Caco-2 (**g**,**h**), HEK 293 (**i**,**j**), and RAW264.7 (**k**,**l**) cells.

**Figure 8 foods-12-04211-f008:**
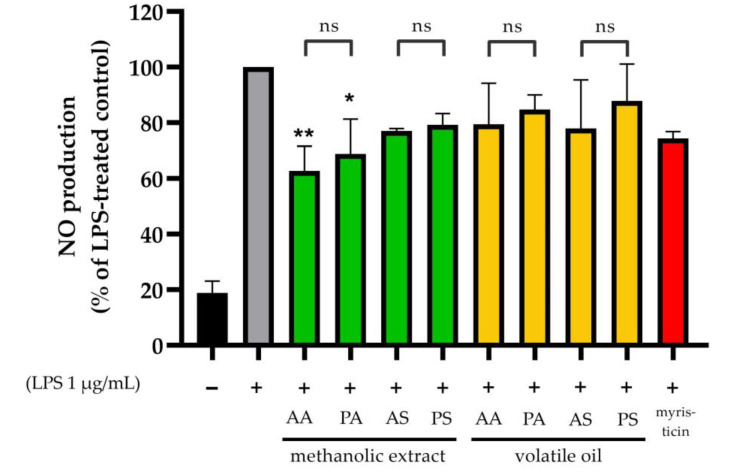
Results of NO production inhibition (%relative to LPS-treated control) for methanolic extracts and volatile oils in RAW 264.7 cells with an IC_20_ or non-toxic concentration. The bar graph illustrates the LPS-untreated group (black), the LPS-treated group (grey), the LPS+methanolic extract of AA, PA, AS, PS-treated group (green), and the LPS+volatile oil of AA, PA, AS, PS-treated group (yellow), and the LPS+myristicin-treated group (red). Each value representes as mean ± SD of three independent experiments. * *p* < 0.033, ** *p* < 0.002 compared to control group, ns means not statistically significant between the groups.

**Table 1 foods-12-04211-t001:** Lists of leaf, aril, and seed samples from authentic *M. fragrans* and commercial nutmeg samples collected in Thailand.

Specimen Code	Sample Typologies	Localities		Dimensions (cm)		
		Regions	Districts, Provinces	Length	Width	Thickness
Leaves						
AL1	Fresh leaves	East	Makham, Chanthaburi	-	-	-
AL2	Fresh leaves	South	Lang Suan, Chumpon	-	-	-
AL3	Fresh leaves	South	Hat Yai, Songkhla	-	-	-
Arils						
Authentic samples						
AA1	Dried arils	East	Makham, Chanthaburi	2.60 ± 0.53	0.90 ± 0.51	0.05 ± 0.01
AA2	Dried arils	South	Lang Suan, Chumpon	3.23 ± 0.60	0.75 ± 0.40	0.06 ± 0.01
AA3	Dried arils	South	Hat Yai, Songkhla	3.10 ± 0.53	1.23 ± 0.76	0.04 ± 0.01
Commercial samples						
PA1	Dried arils	South	Ron Phibun, Nakhon Si Thammarat	2.73 ± 0.75	0.70 ± 0.44	0.04 ± 0.01
PA2	Dried arils	South	Mueang, Nakhon Si Thammarat	2.77 ± 0.68	1.37 ± 0.68	0.04 ± 0.01
PA3	Dried arils	South	Mueang, Phang Nga	2.43 ± 0.20	0.88 ± 0.49	0.05 ± 0.01
PA4	Dried arils	Central	Samphanthawong, Bangkok	3.95 ± 1.18	1.28 ± 0.19	0.06 ± 0.00
PA5	Aril Powder	North	Mueang, Chiang Mai	-	-	-
PA6	Dried arils	South	Mueang, Chumphon	2.63 ± 0.51	0.98 ± 0.49	0.05 ± 0.01
PA7	Dried arils	North	Mueang, Chiang Mai	3.45 ± 0.88	1.03 ± 0.38	0.06 ± 0.01
PA8	Dried arils	North	Mueang, Chiang Mai	3.68 ± 0.36	0.93 ± 0.23	0.06 ± 0.01
PA9	Dried arils	North	Mueang, Chiang Mai	4.80 ± 0.18	1.00 ± 0.46	0.06 ± 0.01
PA10	Dried arils	North	Mueang, Chiang Mai	3.13 ± 0.55	0.87 ± 0.55	0.05 ± 0.00
Seeds						
Authentic samples						
AS1	Dried seeds	East	Makham, Chanthaburi	1.78 ± 0.48	1.40 ± 0.10	-
AS2	Dried seeds	South	Lang Suan, Chumpon	2.00 ± 0.30	1.45 ± 0.05	-
AS3	Dried seeds	South	Hat Yai, Songkhla	2.25 ± 0.22	1.57 ± 0.06	-
Commercial samples						
PS1	Dried seeds	South	Mueang, Nakhon Si Thammarat	1.90 ± 0.37	1.57 ± 0.06	-
PS2	Dried seeds	South	Ron Phibun, Nakhon Si Thammarat	2.05 ± 0.57	1.60 ± 0.10	-
PS3	Dried seeds	Central	Samphanthawong, Bangkok	3.78 ± 0.19	1.78 ± 0.14	-
PS4	Dried seeds	South	Mueang, Chumphon	2.03 ± 0.25	1.47 ± 0.03	-
PS5	Dried seeds	Central	Samphanthawong, Bangkok	3.07 ± 0.25	1.60 ± 0.10	-
PS6	Seed Powders	North	Mueang, Chiang Mai	-	-	-
PS7	Seed Powders	North	Mueang, Chiang Mai	-	-	-
PS8	Seed Powders	North	Mueang, Chiang Mai	-	-	-
PS9	Dried seeds	North	Mueang, Chiang Mai	2.38 ± 0.28	1.65 ± 0.05	-
PS10	Dried seeds	North	Mueang, Chiang Mai	3.03 ± 0.25	1.57 ± 0.15	-
PS11	Dried seeds	North	Mueang, Chiang Mai	2.20 ± 0.10	1.23 ± 0.06	-
PS12	Dried seeds	North	Mueang, Chiang Mai	2.92 ± 0.14	1.65 ± 0.05	-

**Table 2 foods-12-04211-t002:** GC-MS results of the major phytochemicals found in the volatile oil extracted from the representative aril and seed of globose and oval-shaped nutmegs.

No.	RT ^a^	Name of the Compound	MW ^b^	MF ^c^	%AUC ^d^			
					AA1	AS1	PA4	PS3
1	6.7	α-pinene	136	C_10_H_16_	15.68	11.74	3.04	3.41
2	7.9	β-phellandrene	136	C_10_H_16_	33.94	30.98	18.99	27.00
3	9.7	(5R)-1-methyl-5-(1-methyl ethenyl)cyclohexene	136	C_10_H_16_	7.43	5.89	20.29	18.71
4	10.8	terpinene	136	C_10_H_16_	3.64	2.91	2.81	2.72
5	15.7	terpinen-4-ol	154	C_10_H_18_O	6.48	4.22	4.34	6.93
6	20.4	safrole	162	C_10_H_10_O_2_	0.32	6.17	33.35	19.61
7	30.1	myristicin	192	C_11_H_12_O_3_	5.23	3.39	0.04	1.48

^a^ Retention time; ^b^ Molecular weight; ^c^ Molecular formula; ^d^ %Area under the curve.

**Table 3 foods-12-04211-t003:** Phytochemical constituents of the methanolic aril and seed extracts of globose and oval-shaped nutmeg samples analyzed using LC-MS/MS in the negative ion mode.

RT (min)	*m*/*z*	MS/MS Fragments	Formula	Tentative Compounds	Peak Intensity (%)			
					Arils		Seeds	
					Globose Shape	Oval Shape	Globose Shape	Oval Shape
1.718	179.056	59.0143, 71.0138	C_6_H_12_O_6_	β-D-Glucose	-	11.49	-	-
1.790	387.1143	179.0551, 341.1089	C_18_H_20_N_4_O_4_S	N-((5-(Dimethylamino)-1-naphthyl)sulphonyl)-L-histidine	-	35.68	-	-
1.794	341.1092	89.0238, 179.0565, 341.1084	C_12_H_22_O_11_	Sucrose	-	10.37	-	-
16.783	740.4916	61.9889, 220.0473, 740.4914	C_40_H_72_NO_9_P	Phosphatidylserine	-	-	-	4.52
16.789	723.5014	250.0523, 677.4946	C_41_H_73_O_8_P	Phosphatidic acid	-	-	-	3.57
19.242	571.2881	152.9956, 255.2328, 571.2876	C_25_H_49_O_12_P	Phosphatidylinositols	-	8.60	-	-
19.695	293.1755	71.0139, 236.1049	C_17_H_26_O_4_	Myrsinone	36.92	5.52	4.63	2.88
20.185	597.3036	112.9851, 281.2470, 597.3029	C_27_H_51_O_12_P	1-Oleoylglycerophosphoinositol	-	6.40	-	-
20.463	471.1635	112.9855, 247.1326, 357.1705	C_25_H_28_O_9_	C1’-C9-Glycosylated UWM6	-	-	5.44	-
20.470	379.1522	269.1146, 379.1517	C_23_H_24_O_5_	8-Desoxygartanin	-	-	1.27	-
20.521	357.1701	109.0293, 247.1332, 357.1696	C_21_H_26_O_5_	Malabaricone C	8.06	-	37.85	49.53
20.870	222.0765	190.0507, 222.0767	C_11_H_13_NO_4_	Methyl o-methoxyhippuric acid	6.02	5.61	-	-
20.947	325.1449	254.0588, 310.1208	C_20_H_22_O_4_	Dehydrodieugenol	-	-	3.41	3.55
21.054	455.1679	68.9959, 112.9855, 341.1750	C_23_H_27_F_3_O_6_	9, 11, or 15-keto Fluprostenol	-	-	3.65	6.08
21.058	404.1713	61.9887, 109.0291, 341.1751	C_21_H_27_NO_7_	Clivorine	-	-	1.45	-
21.075	341.1754	109.0296, 231.1393, 341.1761	C_21_H_26_O_4_	Neotriptophenolide	-	-	37.88	27.53
21.224	239.0671	151.0769, 239.0675	C_10_H_12_N_2_O_5_	(±)-2-(1-Methylpropyl)-4,6-dinitrophenol	31.46	8.73	-	-
21.241	295.2281	123.1166, 171.1017, 277.2188	C_18_H_32_O_3_	Dimorphecolic acid	7.46	-	-	-
21.785	713.3358	395.1865, 519.2407, 713.3349	C_35_H_54_O_15_	Cheirotoxol	-	-	3.38	-
27.741	154.9738	110.9843, 154.9742	C_2_H_5_O_6_P	Phosphoglycolic acid	10.08	7.60	1.04	2.34

**Table 4 foods-12-04211-t004:** BLAST-based similarity identification results for authentic *M. fragrans*, globose nutmeg, and oval-shaped nutmeg, across five DNA barcoding regions.

DNA Barcode Regions	BLAST Results				
	Matched Species	Accession Number	%Query Cover	%Identity	Max Score
Authentic *M. fragrans*					
ITS	N/A	N/A	N/A	N/A	N/A
*mat*K	*M. fragrans*	KT445278	100	100.00	1498
	*M. teysmannii*	NC_079584	100	100.00	1498
	*M. argentea*	OP866724	100	100.00	1498
*rbc*L	*M. fragrans*	MH069804	100	100.00	968
	*M. malabarica*	KY945260	100	100.00	968
	*M. inners*	MG817056	100	100.00	968
*trn*H-*psb*A	*M. fragrans*	NC_060715	100	99.44	652
	*M. fragrans*	LC461928	100	98.89	640
	*M. yunnanensis*	NC_060716	100	97.49	612
	*M. teysmannii*	NC_079584	100	97.21	606
*trn*L-F	*M. fragrans*	NC_060715	100	100.00	1332
	*M. teysmannii*	NC_079584	100	100.00	1332
	*M. argentea*	OP866724	100	99.86	1327
Globose nutmeg					
*trn*H-*psb*A	*M. fragrans*	KX675160	90	100.00	601
	*M. fragrans*	OK052841	96	99.71	641
	*M. fragrans*	MF802872	91	99.70	601
Oval-shaped nutmeg					
*trn*H-*psb*A	*M. argentea*	OP866724	100	100.00	713
	*M. teysmannii*	NC_079584	100	98.97	691
	*M. yunnanensis*	NC_060716	100	98.46	680

N/A means data not available.

**Table 5 foods-12-04211-t005:** IC_20_ and IC_50_ values (μg/mL) of the methanolic extracts and volatile oils against RAW 264.7 and HaCaT cells for 24 h.

Extract	Sample Code	RAW 264.7		HaCaT	
		IC_20_	IC_50_	IC_20_	IC_50_
Methanolic extract	AA1	14.8 ± 4.6	59.2 ± 18.3	13.4 ± 0.7	53.5 ± 2.7
	AS1	44.6 ± 20.5	178.2 ± 82.1	20.3 ± 0.9	81.1 ± 3.8
	PA4	40.7 ± 4.8	162.8 ± 19.1	12.9 ± 0.5	51.4 ± 2.1
	PS3	31.0 ± 5.7	124.1 ± 22.9	6.6 ± 0.2	26.3 ± 0.7
Volatile oil	AA1	>100	>100	>100	>100
	AS1	>100	>100	>100	>100
	PA4	>100	>100	>100	>100
	PS3	>100	>100	>100	>100
Myristicin standard		>200	>200	>100	>100

**Table 6 foods-12-04211-t006:** IC_20_ and IC_50_ values (μg/mL) of the methanolic extracts and volatile oils against 3T3-L1, Caco-2, HEK 293, and RAW 264.7 cells for 48 h.

Extract	Sample Code	3T3-L1		Caco-2		HEK 293		RAW 264.7	
		IC_20_	IC_50_	IC_20_	IC_50_	IC_20_	IC_50_	IC_20_	IC_50_
Methanolic extract	AA1	25.3 ± 11.2	101.3 ± 44.7	10.3 ± 1.8	41.2 ± 7.2	20.1 ± 8.6	80.4 ± 34.5	8.2 ± 1.9	32.7 ± 7.5
	AS1	30.0 ± 11.8	120.1 ± 47.2	10.9 ± 4.3	43.4 ± 17.4	23.4 ± 4.0	93.7 ± 16.0	9.7 ± 0.9	38.9 ± 3.6
	PA4	56.0 ± 15.9	223.9 ± 63.8	26.7 ± 3.9	106.6 ± 15.6	42.7 ± 5.8	170.9 ± 23.1	30.1 ± 8.9	120.3 ± 35.5
	PS3	54.1 ± 21.3	216.4 ± 85.2	17.4 ± 3.2	69.7 ± 12.8	26.0 ± 1.3	103.8 ± 5.2	16.6 ± 5.0	66.4 ± 19.9
Volatile oil	AA1	67.9 ± 18.7	271.5 ± 74.8	32.0 ± 1.1	128.2 ± 4.5	48.1 ± 5.8	192.4 ± 23.2	33.7 ± 10.8	134.8 ± 43.0
	AS1	76.8 ± 10.7	307.0 ± 42.7	41.0 ± 9.0	163.8 ± 36.1	58.5 ± 20.4	234.0 ± 81.6	38.4 ± 5.2	153.5 ± 20.8
	PA4	63.9 ± 10.5	255.4 ± 42.1	49.4 ± 6.8	197.7 ± 27.1	53.1 ± 5.2	212.6 ± 21.0	44.5 ± 4.9	177.8 ± 19.6
	PS3	69.9 ± 12.1	279.7 ± 48.5	35.4 ± 3.8	141.6 ± 15.1	52.8 ± 12.9	211.0 ± 51.7	43.2 ± 8.7	172.7 ± 34.8

## Data Availability

Data are contained within the article and Appendix A.

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
