# Peer review of "The Identification and Cytotoxic Evaluation of Nutmeg (Myristica fragrans Houtt.) and Its Substituents"

_foods, 2023, doi:10.3390/foods12234211_

Round 1

Reviewer 1 Report

Comments and Suggestions for Authors This study examined two morphological variations of nutmeg found in the Thai market, the globose and oval shapes, focusing on their botanical origins, phytochemical composition, DNA barcoding, cytotoxicity, and anti-nitric oxide production. The results confirmed that the globose-shaped nutmeg is M. fragrans, while the oval-shaped variant is M. argentea. Cytotoxicity tests showed similar profiles between the two, suggesting that blending or substituting them in products maintains therapeutic integrity without safety concerns. The content of this work is solid; however, the major problem lies in the writing and organization of the paper, which hinders its scientific value. Reading it feels like entering a maze rather than navigating a well-structured grand temple. It seemingly giving us less scientific interest, since the conclusion of M. fragrans and M. argentea appears to be safe and does not compromise their therapeutic potential. So claim any of them as "Adulterant" from this point of view may be problematic, not only from scientifc but from logicial point of view. Anyway, revisions should be made before the true scientific standingpoint of the authors being justified. I feel another round of revision is necessary after the writing improved. To improve the overall writing, I suggest the following:

1. It's important to ensure that the abstract and conclusion sections of your paper are crystal clear and concise. In the abstract, try to state your primary research question or hypothesis in a straightforward manner. In the conclusion, summarize your findings succinctly, and emphasize why they matter in practical terms. This will help readers quickly grasp the essence of your study.

2. Introduction: When it comes to the introduction, be explicit about the research gap or the specific questions your study aims to address. Organize your introduction in a way that guides the reader smoothly through the background information. To enhance readability, consider separating different aspects of the introduction into distinct paragraphs, and try to keep it concise by trimming unnecessary details.

3. In the materials and methods section, clarity is paramount. Simplify your descriptions of sample collection and preparation, ensuring that essential details are explained without overwhelming the reader. Offer context for your choice of extraction methods, clarifying why you opted for specific ones. Visual aids, like diagrams or flowcharts, can enhance clarity, and consistent numbering and labeling will make the information more accessible. If you mention statistical analyses, briefly explain why they were performed and what they aimed to reveal.

4. In the disscussion and conclusion, begin your conclusion by summarizing the most significant findings of your study. Give readers a brief overview before diving into the details. This summary will serve as a quick reference point for your audience. Also, practical implications in conclusion is important. When concluding your paper, go beyond just summarizing your findings. Highlight the practical implications of your research. Explain how your results can benefit the industry, consumers, or regulatory bodies. Additionally, suggest possible directions for future research to guide others in the field.

5. the DNA Barcoding methodology perhaps is one of the highlight of this study, however it is not very clear. In the conclusion, provide a clear yet concise summary of your DNA barcoding methodology. Explain why you chose the trnH-psbA region and how it helped in distinguishing between nutmeg samples. Consider including a simple visual aid, like a flowchart, to illustrate the DNA barcoding process. Finally, discuss the significance of your BLAST and neighbor-joining tree analyses. Comments on the Quality of English Language Also, overall the text is mostly clear, but some/many sentences could be made more concise for better readability. Here are suggestions of minor grammatical errors, and the text, taking part of the introduction as example:

Line 50: "named of the spice" should be "named the spice." Line 52: "by the dried aril of the ripen fruit" should be "by the dried aril of the ripe fruit." Line 56: "pharmaceutical industries such as perfume" should be "pharmaceutical industries, such as perfume." Line 66: "are useful for atherosclerosis [12]." could be rephrased for clarity, such as "have potential therapeutic applications, particularly in addressing atherosclerosis [12]." Line 80: "Among a ton of production" should be "Among a ton of production," Line 86: "discerning of processed" should be "discerning processed." Line 106: "nutmeg plant is mainly cultivated" could be more concise as "nutmeg is primarily cultivated." Line 108: "applied for making savory dishes" could be rephrased as "used in savory dishes." Line 112: "the different among the two shapes" should be "differences between the two shapes." Line 113: "such as DNA barcode" could be improved by saying "including DNA barcoding."

Reviewer 2 Report

Comments and Suggestions for Authors

Overall, the work is interesting and well-presented; however, there are some aspects that, in my opinion, need clarification:

  1. Volatile oil was obtained through distillation. The process should be detailed more thoroughly.
  2. The extracted volatile oil for GC-MS analysis was used, but the sample treatment for GC-MS analysis is not adequately explained. Please provide further details.
  3. The chromatograms do not provide additional information, and the images are unclear and not easily readable. I recommend not including them in the main text.
  4. Since fragmentation may vary with source conditions, for fragments identified through MS/MS mode, references used for substance identification should be indicated (with citations). Additionally, precise source fragmentation conditions for each substance must be provided.
  5. Figure 7 is difficult to read; please improve its clarity.
  6. The conclusions are lacking,  and, in my opinion, they should be expanded to include the implications of the work.

Reviewer 3 Report

Comments and Suggestions for Authors

The authors evaluated the "Identification and Cytotoxic Evaluation of Nutmeg (Myristica fragrans Houtt.) and Its Adulterants"

The paper is well-written and well-organized.

Some minor remarks are follow

Please about abstract write a single paragraph of about 200 words maximum

Line 84.Merr. & L.M.Perry?

Round 2

Reviewer 1 Report

Comments and Suggestions for Authors In the revised manuscript, the overall writing has improved as recommended, making the authors' scientific standpoint clearer. However, I believe the work still lacks focus despite the clear evidence of extensive analysis performed to identify differences between the globose and oval shapes. Below are some problems:
1. Among the series of tests conducted by the authors, it remains unclear which is most important. If a single test can identify nutmeg, why were so many tests performed without explaining the advantages and disadvantages within the context of your research goal? Providing this information would better inform potential readers, including government officials and researchers, about the most effective way to identify nutmeg.
2. Furthermore, the use of the term "Adulterant" remains problematic and illogical. Consider using terms such as "alternative plant materials" or similar. The flow of the study should be to identify the differences and whether these differences are harmful. If they are harmful, "adulterant" would be appropriate. Otherwise, the study's logic is conflicting and confusing.
3. Regarding HPLC, why are there so many peaks (Figure 5) when only one is identified? If the goal is simply to compare myristicin, TLC would suffice. There is no need to report any HPLC data, especially since LC-MS is included later, which may render HPLC redundant. The manner in which the HPLC and chromatogram analysis is presented is unusual for its lack of compound identification.
4. For LC-MS, why are there no compounds reported between 1.8 and 16.7 min? This is highly unusual for compound identification, suggesting that you may need to adjust the gradient or that you've overlooked numerous informative peaks. What do the TIC or BPI look like in your raw data?
5. Why are the LC-MS and HPLC chromatogram parameters different? They should be the same for a side-by-side comparison.
6. Table 2. the % height is not a usual way to express relatative concentration. Use absolute or relative intergrated peak area instead.
7. Figure 6. is very small. Use a table instead for a clear, detailed data of GC-MS peaks.
8. There are some occurances of "Mass spectroscopy" in the manuscipt. "Mass spectroscopy" should be "Mass spectrometry".

Reviewer 2 Report

Comments and Suggestions for Authors

The article has significantly improved. However, I would like further clarification. Regarding Point 2: The extracted volatile oil was used for GC-MS analysis, but the sample treatment for GC-MS analysis is not adequately explained. The authors have added the following sentence: 'Oil samples (500 ppm) were prepared in absolute methanol, filtered through 0.45 μm syringe filters, and transferred into 2 mL glass vial bottles.' From this sentence, it appears that the methanolic extract is being injected. How do the authors justify this choice? Typically, methanol is not a recommended choice for GC determinations. If I have misunderstood, please provide clarification . 

Furthermore, I believe that, in addition to indicating the reference database, the authors should specify the source fragmentation conditions adopted for the identification of each analyte. As I had already pointed out in my previous review, the identification of fragments may vary with fragmentation source conditions
